# Extended Interactions between HIV-1 Viral RNA and tRNA^Lys3^ Are Important to Maintain Viral RNA Integrity

**DOI:** 10.3390/ijms22010058

**Published:** 2020-12-23

**Authors:** Thomas Gremminger, Zhenwei Song, Juan Ji, Avery Foster, Kexin Weng, Xiao Heng

**Affiliations:** Department of Biochemistry, University of Missouri, Columbia, MO 65211, USA; tjgyt7@mail.missouri.edu (T.G.); songzh@missouri.edu (Z.S.); JiJ@missouri.edu (J.J.); alfb28@mail.missouri.edu (A.F.); kwtf8@mail.missouri.edu (K.W.)

**Keywords:** HIV-1 PBS, tRNA^Lys3^, reverse transcription, degradation, genomic RNA integrity

## Abstract

The reverse transcription of the human immunodeficiency virus 1 (HIV-1) initiates upon annealing of the 3′-18-nt of tRNA^Lys3^ onto the primer binding site (PBS) in viral RNA (vRNA). Additional intermolecular interactions between tRNA^Lys3^ and vRNA have been reported, but their functions remain unclear. Here, we show that abolishing one potential interaction, the A-rich loop: tRNA^Lys3^ anticodon interaction in the HIV-1 MAL strain, led to a decrease in viral infectivity and reduced the synthesis of reverse transcription products in newly infected cells. In vitro biophysical and functional experiments revealed that disruption of the extended interaction resulted in an increased affinity for reverse transcriptase (RT) and enhanced primer extension efficiency. In the absence of deoxyribose nucleoside triphosphates (dNTPs), vRNA was degraded by the RNaseH activity of RT, and the degradation rate was slower in the complex with the extended interaction. Consistently, the loss of vRNA integrity was detected in virions containing A-rich loop mutations. Similar results were observed in the HIV-1 NL4.3 strain, and we show that the nucleocapsid (NC) protein is necessary to promote the extended vRNA: tRNA^Lys3^ interactions in vitro. In summary, our data revealed that the additional intermolecular interaction between tRNA^Lys3^ and vRNA is likely a conserved mechanism among various HIV-1 strains and protects the vRNA from RNaseH degradation in mature virions.

## 1. Introduction

One of the most distinctive features of retroviruses is that their RNA genome is reverse-transcribed into double-stranded DNA [1], which is integrated into the host genome during infection [2]. Catalyzed by viral reverse transcriptase (RT) [3], the human immunodeficiency virus 1 (HIV-1) reverse transcription initiates on a double-stranded region where the 3′-end of tRNA^Lys3^ forms 18 base pairs with complementary residues in the primer binding site (PBS) (Figure 1a–c) [2,3]. It has been reported that extended interactions between tRNA^Lys3^ and viral RNA (vRNA) are necessary for efficient reverse transcription and tRNA^Lys3^ selection [4,5,6,7,8]. The extended interactions are believed to maintain the overall tertiary structure of the RNA complex to efficiently load the RT and facilitate the transition from reverse transcription initiation to elongation [7,8]. Attempts to force HIV-1 to use alternative tRNAs have been challenging, as HIV-1 RNA carrying a mutated PBS complementary to alternative tRNAs co-packaged in virions exhibited a delay in viral replication, and the PBS residues eventually reverted back to wild-type (WT) sequences. In order to prevent reversion, additional mutations in the A-rich loop region of the vRNA were introduced to be complementary to the anticodon residues of alternative tRNAs, supporting the role of the A-rich loop in the selection of tRNA^Lys3^ [4,5,9,10,11]. The A-rich loop residues are highly conserved in HIV-1 [12,13]. A similar bulge is also present in other retroviruses, including HIV-2 and simian immunodeficiency virus (SIV) and exists at a similar distance to the PBS [12,14].

Numerous efforts have been spent to identify the extended interactions between vRNA and tRNA^Lys3^ in the annealed complex. However, based on different types of experimental systems and different viral strains, there are conflicting models about the exact nature of these extended interactions. In vitro and in situ probing of the annealed RNA complex of HIV-1 subtype A revealed that the A-rich loop residues of the vRNA formed base pairs with residues in the anticodon and anticodon stem of tRNA^Lys3^ [12,15], and cell-free reverse transcription assays reported that such interactions facilitated the efficient transition from initiation to elongation during reverse transcription [7,8]. However, NMR studies of the annealed complex did not detect the A-rich loop: anticodon interaction [16,17,18]. In HIV-1 subtype B, it was reported in the tRNA selection assays that additional mutations in the A-rich loop region were necessary to maintain the usage of non-self tRNAs during virus replication, supporting the role of the A-rich loop in tRNA selection [4,9,19,20,21]. On the other hand, in vitro reverse transcription assays and biophysical studies of the annealed complex suggest an alternative model, in which the primer activation signal (PAS) residues upstream of the PBS form intermolecular base pairs with TΨC residues in tRNA^Lys3^ [22,23,24]. The reverse transcription assays suggested that the PAS helped to promote the efficient initiation of reverse transcription [22,24,25]. However, such an interaction was not detected in chemical probing and was not supported due to the lack of an adaptive PAS mutation in long-term virus replication [26]. Additionally, a structural analysis by cryo-electron microscopy did not report any extended interactions in HIV-1 subtype B RNA [27], suggesting that these extended interactions are likely to be disrupted when cDNA is actively synthesized. Despite their prominent roles in reverse transcription, molecular details of the extended interactions in the vRNA: tRNA^Lys3^ complex remain unclear.

Here, we present NMR evidence supporting the A-rich loop: tRNA^Lys3^ anticodon interaction. Consistent with previous studies [8], abolishing such an interaction led to a decrease in infectivity and reduced the synthesis of reverse transcription products. Our in vitro primer extension experiments revealed that disruption of the A-rich loop: tRNA^Lys3^ anticodon interaction in HIV-1 subtype A RNA resulted in an increased affinity for RT and enhanced the reverse transcription efficiency. In the absence of deoxyribose nucleoside triphosphates (dNTPs), the RNaseH activity of RT degraded the annealed vRNA: tRNA^Lys3^ complex, and the complex with a disrupted A-rich loop: tRNA^Lys3^ anticodon interaction underwent an accelerated degradation. Consistently, the loss of vRNA integrity was observed in virions carrying A-rich loop mutations. We also reported similar results in HIV-1 subtype B RNA, that the A-rich loop: anticodon interaction affected the RT-binding affinity, primer extension efficiency, and RNA integrity. Additionally, we show that the nucleocapsid (NC) protein was required to promote the extended vRNA: tRNA^Lys3^ interaction under in vitro conditions. Together, our results suggest a conserved mechanism among various HIV-1 strains: that the extended vRNA: tRNA^Lys3^ interaction protects the vRNA in the mature virions prior to entering a new host cell.

## 2. Results

### 2.1. A-Rich Loop Mutations in the PBS Segment Disrupted Extended Interaction Between vRNA and tRNA^Lys3^ and Led to Reduced Infectivity

In our study, a minimal HIV-1 MAL construct PBS segment (nt 123−240, with two non-native G-C pairs) was synthesized by in vitro T7 transcription to investigate the possible extended interactions between vRNA and tRNA^Lys3^ (Figure 1a). The NMR assignment of residues in the A-rich loop hairpin and the PAS stem were made by referencing the spectra with the NMR data collected for the control RNA fragments (Appendix A). To test the hypothesis that the A-rich loop is involved in extended vRNA: tRNA^Lys3^ interactions, G162-A167 in the PBS segment were mutated to a UNCG tetraloop enclosed by a CG base pair (PBS-MutA, Figure 1b). The wild-type (WT) PBS segment and PBS-MutA were annealed with tRNA^Lys3^ (Figure 1c) by incubating the mixture at 94 °C for 5 min and slowly cooling down to room temperature (Materials and Methods), and the predicted models are shown in Figure 1d–f. The annealed complexes were examined by an electrophoretic mobility shift assay (EMSA) (Figure 1g), and the results show that both the WT PBS segment and PBS-MutA formed a complex with tRNA^Lys3^. The A-rich loop mutation did not disrupt the formation of the PBS-MutA: tRNA^Lys3^ complex, suggesting that the A-rich loop: anticodon interaction is not the major inter-molecular interaction contributing to the formation of the vRNA: tRNA^Lys3^ complex. Next, NMR data were collected for the PBS segment:-tRNA^Lys3^ and PBS-MutA: tRNA^Lys3^ complexes. The annealing of tRNA^Lys3^ to the WT PBS segment led to a drastic spectral change (Appendix A). Spectra broadening was observed in the PBS segment: tRNA^Lys3^ complex because of the large molecular size (63 kDa). Several characteristic peaks belonging to the stem of the A-rich loop hairpin disappeared or shifted upon tRNA annealing. In particular, A155.H2 and A174.H2 gave rise to characteristic up-field shifted chemical shifts in the PBS segment RNA, and they disappeared upon tRNA^Lys3^ annealing (Appendix A and Figure 1h). The NMR spectral changes suggest that the base pairings in the stem were altered after tRNA^Lys3^ annealing (Figure 1b). The mutations in the PBS-MutA did not affect the A-rich loop structure in the RNA, as both the A174 and A155 peaks appeared at the same place. The annealing of tRNA^Lys3^ to the PBS-MutA segment did not lead to changes of chemical shifts of residues in the A-rich loop hairpin, and the A155.H2 and A174.H2 remained unchanged (Figure 1h). NMR studies of the PBS segment RNA and tRNA^Lys3^-annealed complexes suggest that the mutations in PBS-MutA successfully disrupted the predicted intermolecular interactions between the vRNA and tRNA^Lys3^.

To test whether the mutations of the A-rich loop affect the virus infectivity, a chimeric molecular clone containing the HIV-1 MAL 5′ untranslated region (UTR) and NL4.3 backbone (Δ5′UTR and Δenv, with a reporter an enhanced green fluorescent protein (EGFP) under a cytomegalovirus (CMV) promoter [28]) was made, and viruses pseudo-typed with glycoprotein G from the vesicular stomatitis virus (VSV-G) were prepared (Figure 2a,b). PBS-MutA mutations were introduced to the corresponding position of the 5′UTR region in the molecular clone to assess their impact on viral infectivity. An equivalent amount of WT and MutA virions were used in single-cycle infectivity assays, in which the infectivity was quantified by measuring the EGFP signals in infected cells. The results show that the mutation of the A-rich loop led to an approximated 60% reduction of infectivity compared to the WT (Figure 2d).

Since the disruption of the extended vRNA: tRNA^Lys3^ interaction is likely to affect RT loading and reverse transcription initiation [7,8], the reverse transcription products of the WT and MutA infection were quantified. Specific PCR primers and a probe were used to measure the amount of reverse transcription cDNA products two hours post-infection [29]. Consistent with the infectivity results, a mutation of the A-rich loop led to a two-fold decrease in reverse transcription products compared to WT (Figure 2e). These data demonstrate that mutations in MutA negatively impacted the early stage of viral replication.

### 2.2. Mutation of the A-Rich Loop Increased the Affinity of RT Binding to the vRNA: tRNA^Lys3^ Complex and Enhanced the Primer Extension Efficiency

To test whether the mutation of the A-rich loop altered the affinity of RT binding to the vRNA: tRNA^Lys3^ complex, we employed microscale thermophoresis (MST) to measure the RT-binding affinity. Since reverse transcription initiates on the genomic RNA after packaging, a 5′UTR construct (nt 1–358) favoring the dimer conformation with a packaging signal exposed was prepared to represent the vRNA [30]. The MutA mutations were introduced to generate 5′UTR-MutA. Both the 5′UTR and 5′UTR-MutA were labeled with a 5′-fluorescein thiosemicarbazide (FTSC) and annealed to tRNA^Lys3^ at 50 nM and a 1:1 ratio. RT was titrated to reach a final concentration ranging from 0.00252 µM to 20.6 µM in the WT complex samples and 0.000763 µM to 6.2 µM in the mutant complex samples. The fluorescence signal changes upon RT titrations versus the RT concentrations that were plotted (Figure 3a,b). An analysis of the MST traces showed that the affinity of RT to the 5′UTR: tRNA^Lys3^ complex is 1356 ± 408 nM (Figure 3a), and the affinity to the 5′UTR-MutA: tRNA^Lys3^ complex is 78.3 ± 27.6 nM (Figure 3b). Disruption of the A-rich loop: anticodon interaction resulted in a nearly 20-fold affinity enhancement of RT binding to the tRNA^Lys*3*^-annealed complex for reverse transcription initiation.

Since the annealed complex differed in structure between the PBS segment and PBS-MutA, and the MutA virus was deficient in producing (-)cDNA in infected cells (Figure 2), we hypothesized that the reverse transcription efficiency is affected. To test whether mutations of the A-rich loop led to an alteration of primer extension efficiency, an in vitro primer extension assay was employed. 5′-Cy3-labeled tRNA^Lys3^ was annealed to WT 5′UTR and 5′UTR-MutA and incubated with dNTPs and RT at 37 °C for various amounts of time. The polymerase interaction generated a cDNA product covalently linked to the Cy3-labeled tRNA^Lys3^ primer, and the fully extended product was expected to be 255 nt long (76 nt (tRNA^Lys3^) +179 nt (cDNA product)). Primer extension products were resolved by polyacrylamide gel electrophoresis and imaged by fluorescent scanning (Figure 3c). It appeared that the mutation of the A-rich loop led to an increased primer extension, as more full-length products were synthesized on the mutant template at each tested time point (Figure 3d). These data agree with the MST affinity results that the 5′UTR-MutA: tRNA^Lys3^ had a stronger affinity to RT, and thus, the reverse transcription efficiency was greater than that of the WT 5′UTR: tRNA^Lys3^ template.

### 2.3. The A-Rich Loop:Anticodon Interaction Protected vRNA from Degradation by RT in the Absence of dNTP

Our cell-based experiments revealed that the MutA-containing virus had a deficiency in replication step(s) before or during reverse transcription. However, the in vitro binding and primer extension results show that the vRNA: tRNA^Lys3^ complex with A-rich loop mutations was favored by RT and exhibited enhanced efficiency in producing cDNA products. Early reverse transcription may take place in mature virions, but the rate is limited by the dNTP concentrations in the extracellular environment [31,32,33,34]. Thus, there might be a window of time that RT can bind to the RNA complex but is unable to synthesize cDNA before the virion enters a host cell. While the RNaseH activity of RT mainly cleaves the RNA on a DNA: RNA duplex, the cleavage of RNA on an RNA: RNA template has been previously reported [35]. Therefore, we hypothesized that, before reverse transcription begins, RT could bind to the RNA complex and degrade the complex with its RNaseH activity. To address this possibility, we designed an in vitro degradation assay to monitor the degradation of the 5′UTR over time in the presence of RT without dNTPs (Figure 4a). The 5′UTR and 5′UTR-MutA were labeled with 3′-FTSC and annealed to tRNA^Lys3^ to visualize the degraded RNA fragments from the 3′-end. The appearance of a degradation band slightly higher than 156 nt was observed in both the WT and mutant but was absent when no RT was added (Figure 4b), indicating the degradation was caused by the addition of RT. The degradation product was stronger in 5′UTR-MutA than in the WT at 40 min and 60 min. We attributed this band to be from the RNaseH activity of RT, as it did not appear when incubating the RNA complex with RT^E478Q^, a RT mutant reported to abolish the RNaseH activity [36,37,38] (Figure 4b).

Since the mutation of the A-rich loop increased the degradation of the 5′UTR in vitro, we then checked the integrity of genomic RNA in virions. According to the in vitro degradation results, RT cleaved at the 3′-end of the PBS and, thus, could generate large fragments that could be detected by PCR primers targeting the RU5 region. Thus, the genomic RNAs extracted from virions were subjected to oligo dT-coated magnetic beads to purify RNA with polyA tails and remove the degraded RNA fragments upstream of the PBS (Figure 4c). qRT-PCR was performed with specific primers/probes targeting RNA regions upstream and downstream of the PBS (RU5 and Gag, respectively). The genomic RNA integrity was calculated as the ratio of RU5: Gag. In each parallel experiment, the RU5:Gag ratio of the MutA virus was always lower than that of the WT, suggesting the genomic RNA in the MutA virus lost its RU5 region in mature virions. Our data show that A-rich loop mutations led to an approximately three-fold reduction of genomic RNA integrity in virions (Figure 4d). Based upon our in vitro results, we attributed this to be from RNaseH-mediated degradation in virions.

### 2.4. The A-Rich Loop: Anticodon Interaction is Conserved in Other HIV-1 Subtypes to Protect vRNA Integrity

To understand if the A-rich loop: anticodon interaction to maintain viral genomic RNA integrity is conserved in other subtypes, we introduced A-rich loop mutations into the subtype B NL4.3 PBS region (PBS-NL4.3) to generate PBSm-NL4.3, in which the unpaired adenosines were substituted by UA pairs (Figure 5a,b). The infectivity of the mutant was measured in single-cycle infectivity assays using the HIV-1 NL4.3-derived reporter virus (pNL4.3-CMV-EGFP Figure 2a) [39]. Pseudo-typed viruses were propagated in 293FT cells, and virus-containing supernatants were harvested 48 h post-transfection. An equivalent amount of WT and mutant viruses were used to infect cells, and the infectivity was analyzed by monitoring the EGFP signals using flow cytometry [39]. As expected, the A-rich loop mutation in the PBSm led to modest infectivity attenuation in the single-round infectivity assay (Figure 5b). Reverse transcription products were quantified two hours post-infection, and the mutations in PBSm led to a two-fold reduction of cDNA synthesis in infected cells (Figure 5c). These data indicate that the A-rich loop mutation in HIV-1 NL4.3 also caused a replication deficiency. To address the hypothesis that the mutation alters the structure of the vRNA: tRNA^Lys3^ complex and, thus, cannot prevent RT from degrading the genomic RNA, the integrity of the genomic RNA extracted from the virions was quantified using the same procedure as illustrated in Figure 4c. As expected, the integrity of the genomic RNA in PBSm viruses were significantly reduced than that of the WT viruses (Figure 5d). Collectively, these cell-based results of the NL4.3 viruses are consistent with the data collected for the MAL strain, suggesting the A-rich loop: anticodon interaction is likely conserved among various HIV-1 strains to protect the genomic RNA integrity in virions prior to entering a new host cell.

### 2.5. Formation of the A-Rich Loop:Anticodon Interaction Requires nucleocapsid (NC) Under In Vitro Conditions

In order to confirm that the A-rich loop: anticodon interaction was disrupted in PBSm-NL4.3, we annealed PBS-NL4.3 and PBSm-NL4.3 with tRNA^Lys3^ in vitro and investigated the complex structure by EMSA and NMR. One of the unexpected findings was that the structure of the annealed complex was different, depending on the annealing conditions. NC-promoted annealing was performed by mixing the PBS-NL4.3 RNA and tRNA^Lys3^ at a 1:1 ratio with various amounts of NC at 37°C. NC proteins were removed from the RNA complex by high salt washes (10 mM Tris-HCl, pH 7.5, 1.5 M NaCl, and 1 mM MgCl_2_), and the annealed complexes were analyzed by EMSA. Under all conditions, a stable vRNA: tRNA^Lys3^ complex was formed in the absence of Mg^2+^ (Figure 6c, top gel). Interestingly, when the complexes were electrophoresed in the presence of 2-mM MgCl_2_, these complexes resolved in different bands (Figure 6c, bottom gel). The complex annealed with one equivalent of NC (NC: PBS-NL4.3: tRNA^Lys3^ = 1:1:1) overnight showed a weak slow-migrating band (slow-complex 1) (Figure 6c, bottom gel, lane 3). The complexes annealed with more NC showed a new slow-migrating band (slow-complex 2). The intensity of these bands increased with the amount of NC they incubated with, even though NC was removed from the RNA samples prior to gel electrophoresis (Figure 6c, bottom gel, lanes 4–7). However, the heat-annealed complex (Materials and Methods) resolved in a similar pattern as the 1-NC-annealed complex (Figure 6c, bottom gel, lane 8). These data suggest an alternative RNA complex structure when more NC was included in the annealing step. We suspected that the 12-NC-annealed complex exposed certain residues that mediate dimerization of the annealed complex (slow-complex 2), but these residues were in a different structure and not available to mediate dimerization when the complex was annealed by 1-NC or by heat. These fast- and slow-migration complex bands were also reported previously in NC-promoted vRNA: tRNA^Lys3^ complexes [40]. Furthermore, the position of the slow complex formed in the 1-NC-annealed complex (slow-complex 1) was lower (migrated faster) than the slow complex formed in the 12-NC-annealed complex (slow-complex 2, Figure 6c), suggesting structural differences in these slow complexes. Further, the NMR analysis of these heat- and NC-annealed complexes were performed to assess their structures. If an A-rich loop: anticodon interaction occurs, then the chemical shifts of residues within or close to the A-rich loop of the PBS-NL4.3 are expected to undergo large shifts. Although we were not able to complete the NMR assignment of the entire complex, we found several characteristic peaks were particularly helpful in distinguishing different RNA complex structures (Figure 6a, b). For example, the A147.H8 signal was detected in the spectra of both heat- and 1-NC-annealed complexes, which were of the same chemical shift and intensity as the A147.H8 signal in the PBS segment spectrum (Figure 6d). These data indicate that the A-rich loop is likely not affected upon tRNA^Lys3^ annealing and support the complex model shown in Figure 6b. However, such a signal was significantly reduced in the 12-NC-annealed complex spectrum, demonstrating that the chemical environment of A147 in the complex annealed by 12 NC was different (Appendix A). We compared the intensity of the 147.H8 signal to the 157.H2, which remained unchanged in all the annealed complexes, and the result showed that approximately 50% of the A147.H8 signals disappeared/shifted upon 12-NC-promoted tRNA annealing (Appendix A). The disappeared or shifted A147 peak supported the extended A-rich loop: anticodon interaction in the model shown in Figure 6a.

We then examined if the mutations in PBSm-NL4.3 disrupted the proposed A-rich loop: anticodon interaction. When annealed by a sufficient amount of NC (NC: RNA = 12: 1), PBSm-NL4.3: tRNA^Lys3^ resolved into two bands that were clearly distinct from the WT complex in a native polyacrylamide gel (Figure 6e). Both the heat-annealed complex and NC-annealed PBSm-NL4.3: tRNA^Lys3^ complex contained a slow-complex 1 shown on the gel, similar to the heat-annealed complex formed by PBS-NL4.3 and tRNA^Lys3^ (Figure 6e). These results suggest that the slow-complex 2 observed in the PBS-NL4.3:tRNA^Lys3^ complex annealed by a sufficient amount of NC is related with the formation of the A-rich loop: anticodon interaction. This is further supported by the NMR data of the PBSm-NL4.3: tRNA^Lys3^ complex annealed by 12 NC. The A147.H8 of PBSm-NL4.3 remained unchanged in the 12-NC-annealed complex (Figure 6f), demonstrating no structural change in or near the A-rich loop region upon tRNA^Lys3^ annealing (Figure 6b).

While the annealing conditions affected the PBS-NL4.3: tRNA^Lys3^ complex structure, we noticed that both the in vitro annealing conditions led to the same complex structure in the MAL PBS segment and 5′UTR (Appendix A). No slow-complex bands were observed when running the heat-annealed or NC-annealed complex on a gel containing Mg^2+^. Additionally, heat annealing was sufficient to disrupt the formation of the A-rich loop hairpin in the PBS segment RNA: tRNA^Lys3^ complex (Figure 1 and Appendix A). These results agree with the previously reported findings that heat and NC annealing resulted in the same complex structure in MAL RNAs [41].

### 2.6. The A-Rich Loop:Anticodon Interaction Protects the Viral RNA from the RNaseH Degradation of RT

Since NC annealing promoted the A-rich loop: anticodon interaction in PBS-NL4.3: tRNA^Lys3^, we next tested if the complex is resistant to RT RNaseH degradation in the absence of dNTPs, as we observed in the MAL RNAs. Both 3′-FTSC-labeled PBS-NL4.3 and PBSm-NL4.3 were annealed to tRNA^Lys3^ in the presence of 12 equivalent NC and then mixed with RT for three hours. While the WT RNA was partially degraded over time, the PBSm-NL4.3 RNA was almost completely degraded after two hours. The major degradation band was slightly higher than the 24-nt RNA ladder, indicating the major cleavage site is at or near C200, the 3′-end of the PBS (Figure 5a). When PBS-NL4.3 was heat-annealed to tRNA^Lys3^, the complex was less resistant to RT degradation, and the degradation rate was similar to that of the PBSm-NL4.3: tRNA^Lys3^ complex (Figure 7b). Collectively, these data demonstrate that the A-rich loop: anticodon interaction protected the viral RNA from RT degradation, and the formation of this extended interaction, in vitro, requires NC. Consistently, RT^E478Q^ did not degrade the template RNA (Figure 7b), suggesting the degradation was mediated by the RNaseH activity of RT.

Primer extension experiments and MST affinity measurements were carried out to examine if the protection of the RNA integrity comes from the reduced affinity of RT binding to the PBS-NL4.3: tRNA^Lys3^ complex. All of the RNA complexes tested in these assays were annealed in the presence of a sufficient amount of NC to promote the formation of the A-rich loop: anticodon base pairs. As shown in Figure 7c, the primer extension efficiency of the WT PBS-NL4.3 was much slower than that of the PBSm-NL4.3, agreeing with the results observed in the MAL RNAs (Figure 3c). An MST analysis of RT binding to the PBSm-NL4.3: tRNA^Lys3^ complex suggests that the A-rich loop mutations led to a stronger affinity (dissociation constants (K_d_) = 46.5 ± 29 nM, Figure 7d, right panel), similar to the RT affinity for PBS-MutA: tRNA^Lys3^ complex (Figure 3b). The MST trace of RT binding to the WT complex annealed by NC exhibited a biphasic binding, that one binding event occurred at low RT concentrations and one occurred at high RT concentrations. Data fitting to a biphasic model show two binding events with K_d1_ = 49.7 ± 9 nM and K_d2_ = 5500 ± 1200 nM (Figure 7d, left panel). According to the NMR studies of the 12-NC-annealed complex, ~50% of the complex did not form the A-rich loop: anticodon interaction (Appendix A). Thus, we think the weak binding is likely RT binding to a vRNA: tRNA^Lys3^ complex with the A-rich loop: anticodon interaction formed, as the affinity is in the same μM range as the affinity of RT for the WT MAL vRNA: tRNA^Lys3^ complex (K_d_ = 1356 ± 408 nM, Figure 3a). The tight binding may come from RT binding to the annealed complex that did not successfully form the A-rich loop: anticodon interaction. Indeed, the affinity of the strong binding event was very close to the affinity measured for both the MAL and NL4.3 RNA complexes containing A-rich loop mutations.

Thus, the in vitro data of NL4.3 RNA confirmed that the A-rich loop: anticodon interaction protected the RNA from RT RNaseH degradation in the absence of dNTP. Our data demonstrate that such extended interactions between vRNA and tRNA^Lys3^ exist in two HIV-1 experimental strains and could be a conserved mechanism that HIV-1 employs to protect its genomic RNA in mature virions prior to entering a newly infected cell.

## 3. Discussion

Our findings demonstrate the importance of the A-rich loop residues upstream of the 18-nt tRNA annealing site on HIV-1 replication. Supported by NMR data, the A-rich loop residues are likely involved in the proposed A-rich loop: anticodon interaction with tRNA^Lys3^ (Figure 1h and Figure 6d). The mutation of the A-rich loop residues in both the MAL and NL4.3 RNAs led to the disruption of the extended interactions. In agreement with previous studies [8], the importance of the A-rich loop and the extended vRNA: tRNA^Lys3^ interactions were demonstrated by a reduced viral infectivity and deficiency in producing cDNA in infected cells (Figure 2 and Figure 5b,c). However, we found RT preferred binding to the A-rich loop mutated vRNA: tRNA^Lys3^ complex and exhibited a higher extension efficiency on the mutant primer: template complex (Figure 3 and Figure 7c,d) than the WT primer: template complex. We therefore hypothesized that the A-rich loop: anticodon interaction could serve to protect the vRNA from degradation by the RNaseH activity of RT in the absence of dNTPs. Using an in vitro degradation assay, we showed that the RT degraded the genome at approximately the 3′-end of the 18-nt tRNA^Lys3^ annealing site. Loss of the segment upstream of the PBS in the viral genomic RNAs carrying A-rich loop mutations were also observed. In summary, our data suggest that the A-rich loop: anticodon interaction serves to protect the viral genome before efficient reverse transcription takes place.

### 3.1. Biophysical Evidence Supports the Extended vRNA: tRNA^Lys3^ Interactions

Decades of effort has been spent to identify the extended vRNA: tRNA^Lys3^ interactions in addition to the 18 base pairings between the PBS and the 3′- of tRNA^Lys3^ and investigate their functions in HIV-1 viral replication. Extended intermolecular interactions between vRNA and tRNA^Lys3^ are important in tRNA selection and reverse transcription initiation [4,5,6,7,8]. Besides the 18-nt annealing base pairs formed between the PBS and tRNA^Lys3^, two major interactions have been proposed, but are yet to be confirmed by further structural evidence. One is between the A-rich loop residues and the anticodon loop of tRNA^Lys3^. Attempts to force HIV to use alternative tRNAs for reverse transcription by mutating the PBS to be complementary to other tRNAs were not successful, as the PBS residues reverted after short-term cultures. However, with additional mutations in the A-rich loop region, the PBS could remain complementary to a non-self tRNA and did not revert back, supporting the role of the A-rich loop in the selection of the use of non-self tRNAs [4,5,42,43]. Liang et. al. showed that deletion of the A-rich loop in HIV-1 HXB2D (Subtype B) led to the reduced infectivity and synthesis of viral DNA. A long-term cell culture led to a partial restoration of the A-rich loop in the A-rich loop-deleted virus [8]. The other interaction is between the PAS residues and the TΨC arm of tRNA^Lys3^, which was supported by cell-free reverse transcription assays and an in vitro biophysical analysis but not by tRNA adaptive selection upon a long-term cell culture [22,23,25,26,44,45,46]. In vitro and in situ probing of the annealed RNA complex of HIV-1 subtype A support the A-rich loop: anticodon model, and cell-free reverse transcription assays demonstrated that such an interaction facilitates the efficient transition from initiation to elongation in reverse transcription [7]. However, NMR studies of the annealed complex were not able to detect the A-rich loop: anticodon interaction [16]. Although the results from different groups and experimental settings are controversial, a consensus has been reached that extended vRNA: tRNA^Lys3^ interactions contribute to tRNA selection and modulate the reverse transcription efficiency and viral infectivity. Here, we present indirect NMR evidence supporting the A-rich loop: anticodon interaction in both MAL and NL4.3 RNAs. The disappearance of the A147.H2 and A155.H2 signals in the MAL PBS segment (Figure 1h) and the reduced signal of A148-H8 in the NL4.3 RNA upon tRNA^Lys3^ annealing (Figure 6d) indicate the residues near the A-rich loop experienced drastic chemical environment changes. The mutating A-rich loop residues in both MAL and NL4.3 RNAs disrupted such extended interactions, as no chemical shift changes of MAL A147.H2, A155.H2, or NL4.3 A147.H8 were observed (Figure 1h and Figure 6f). The mutations in this study were not designed to investigate the possible PAS: TΨC interaction, and the biological function of the PAS: TΨC interaction, if it exists, remains to be investigated.

### 3.2. The A-Rich Loop:Anticodon Interactions Protect Viral Genomic RNA as a Template for Reverse Transcription

The Marquet group introduced similar A-rich loop mutations into a MAL 5′UTR (nt 1–311) substituting GUAAAA (nt 162–167) by CUAUG and used the RNA as a template for tRNA^Lys3^ annealing and primer extension. They reported the mutation had a slower rate of converting from the initiation phase into the elongation phase [7], and the mutant template had a slightly weaker affinity of RT binding [46]. Within the timeframe of our primer extension assays, we did observe the accumulation of reverse transcription products using the 5′UTR-MutA template, while such early pause bands were barely detected when using the WT 5′UTR template (Figure 3c). The discrepancy is likely caused by the low gel resolution in our studies that were not sufficient to distinguish +3 and +5 products from the un-extended primers. Nevertheless, we observed more full-length product accumulation on the mutant template, indicative of efficient primer extension. Deletion of the A-rich loop in the viral RNA of the HXB2 strain was reported to impair reverse transcription initiation, demonstrated by reduced levels of (-)cDNA products in both cell-based and cell-free assays [8]. Our data agreed that the A-rich loop is crucial for viral replication, as MutA viruses were deficient in producing cDNA in infected cells (Figure 2). The discrepancies in the RT-binding affinity and apparent primer extension rate of the A-rich loop mutants are likely caused by the different experimental systems and conditions employed. The structure of the MAL 5′UTR RNA used in our in vitro assays was recently investigated by NMR and was demonstrated to adopt a dimer conformation with NC-binding sites exposed to direct viral genome packaging [30]. Thus, the PBS segment in the 5′UTR is believed to represent, or close to, its conformation in nascent virions when the genomic RNA is packaged. Our NMR data support the formation of an A-rich loop: anticodon interaction in the WT annealed complex and show that such an interaction was disrupted by A-rich loop mutations in MutA (Figure 1h). Using such a system, a stronger RT-binding affinity, enhanced primer extension efficiency, and accelerated degradation were observed in MutA. These results agree with our cell-based assays, where the MutA viruses were deficient in synthesizing cDNA products, because their RNA genome was partially degraded in mature virions. Furthermore, similar phenotypes were observed in NL4.3 when its A-rich loop was deleted (Figure 5), demonstrating the genome RNA protection strategy by the extended A-rich loop: anticodon interaction is conserved among various HIV-1 strains.

### 3.3. NC Annealing and Heat Annealing Lead to Differently Folded RNA Complexes In Vitro

Annealing of tRNA^Lys3^ onto the PBS segment does not occur spontaneously at a physiological temperature, as both the PBS segment and tRNA^Lys3^ are highly structured. Complete annealing requires mature NC, which is produced after Gag is processed by viral protease, as in virio SHAPE activities of the viral RNA from protease-deficient virions did not show a similar pattern of complete tRNA^Lys3^ annealing as that of the WT virions [47]. A recent NMR study uncovered that Murine Leukemia Virus (MLV) NC mediates primer annealing by targeting several regions of both the PBS segment and tRNA^pro^ primer to expose their complementary sequences for annealing [48], emphasizing the importance of NC in promoting tRNA annealing. Our data show that, for the MAL PBS segment: tRNA^Lys3^ complex, the extended A-rich loop: tRNA anticodon interaction is formed in both heat- and NC-annealing conditions. This is consistent with the previously reported results that examined the structure of the MAL PBS segment: vRNA complexes annealed by heat and by NC using enzymatic probing [41]. However, in the case of subtype B RNA, NC is necessary to promote the extended A-rich loop: tRNA anticodon interaction in vitro. Heat can anneal tRNA^Lys3^ onto the PBS-NL4.3, but both the EMSA and NMR data indicate that the structure of the complex was not the same as the NC-annealed complex (Figure 6). The heat-annealed complex was not resistant to RT degradation, which was similar to the complex formed between PBSm-NL4.3 and tRNA^Lys3^ (Figure 7b). These findings are in-line with the primer extension results reported by Liang et. al, that the extension efficiency of the del-A mutant was similar to the WT, because the tRNA^Lys3^ was placed on the PBS by heat [8].

Under in vitro conditions, NC did not fully promote the formation of the vRNA: tRNA^Lys3^ complex with the extended A-rich-loop: anticodon interactions. In Figure 6c, annealing by 12 NC led to the formation of a major band of slow-complex 2 with sufficient time (three hours, lane 7). Annealing with a shorter time (0.3 h, lane 6) led to a mixture of slow-complex 2 and fast-complex. The appearance of multiple bands in the fast complex in lane 6 may be an intermediate that will form slow-complex 2 with additional time. These results demonstrate the slow kinetics of the NC-promoted annealing process. Our NMR results show that, even at a high NC: RNA ratio (12: 1), NC did not fully promote the formation of the vRNA: tRNA**^Lys3^** complex with the extended A-rich loop: anticodon interactions in test tubes (Appendix A). This could be caused by the extensive wash step to remove NC that partially affected the RNA structures. However, a recent single-molecule Förster resonance energy transfer (smFRET) also detected two distinct conformers [18] and suggested a dynamic interplay between the RT and the vRNA: tRNA^Lys3^ complex [49]. While the A-rich loop: anticodon interaction was not detected in the CryoEM structure of the reverse transcription initiation complex [27], such a structure could be a transient structure that temporally protects the genomic RNA integrity.

### 3.4. Rediscovery of the RNaseH Activity of RT

It has been reported that, in addition to the cleavage of RNA in a RNA/DNA hybrid, the RNaseH domain of RT has the ability to cleave RNA in a RNA/RNA hybrid [36,50]. One early report contributed this to a contamination of *Escherichia coli* RNaseIII [51] from a recombinant protein expression. However, it was shown that the mutation of one of the residues required for binding of the divalent metal ions in the RNaseH active site of RT, E478Q, eliminated the cleavage of both RNA in an RNA/RNA hybrid and RNA in a DNA/RNA hybrid [36], suggesting that this activity is inherent to RNaseH. In our in vitro degradation assay, we employed this RT mutation strategy to show that cleavage in the vRNA: tRNA^Lys3^ complex is RNaseH-specific. In both WT and MutA complexes, the degradation bands were eliminated when mixed with RT^E478Q^, suggesting that the degradation was RNaseH-specific. It has also been shown that, in arrested complexes, RNaseH can degrade RNA in an RNA/RNA hybrid 18-bp upstream from the 3′-end of the primer [35], due to the proximity between the RNaseH domain of RT and the 3′-end of the PBS sequence demonstrated in the smFRET studies [49,52]. This cleavage is consistent with the size of our cleavage products in our in vitro degradation assay in both the MAL and NL4.3 RNAs (Figure 4b and Figure 7a).

### 3.5. Affinity of RT to DNA/RNA, DNA/DNA, and RNA/RNA:A-Rich Loop Interaction Creates Hindrance for RT to Bind

The affinity of RT to RNA/RNA is much weaker than to DNA/RNA and DNA/DNA, with disassociation constants (K_d_) of approximately 90 nM for RNA/RNA and 5 nM for DNA/RNA and DNA/DNA [53,54]. Using MST, we observed a similar K_d_ in the mutant vRNA: tRNA^Lys3^ complexes carrying A-rich loop mutations (Figure 3b and Figure 7d). However, the K_d_ observed in MAL WT vRNA: tRNA^Lys3^ was approximately 20-fold weaker. These observations suggest that the A-rich loop: anticodon interaction possibly creates a steric hindrance for RT to bind. The NC-annealed NL4.3 vRNA: tRNA^Lys3^ complex partially formed the A-rich loop: anticodon interaction, and thus, both the tight binding and weak binding were detected (Figure 7d). The RT affinity difference between the WT and MutA complexes explains the differences in observed reverse transcription efficiency, in which the greater complex binding affinity to the mutant led to efficient primer extension. As a result, the weaker complex binding affinity of RT to the WT RNA limited the chance of vRNA degradation by the RNaseH activity of RT. The biophysical data, as well as the in vitro functional results, support the hypothesis that the extended A-rich loop: anticodon interaction serves to protect the vRNA in the absence of reverse transcription and prevents it from being degraded by the RNaseH activity of RT.

## 4. Materials and Methods

### 4.1. Plasmids

The MAL [55] 5′UTR plasmid was a kind gift from the Michael F. Summers Laboratory (University of Maryland, Baltimore, MD, USA). Multiple DNA fragment assembly using the primers listed in Appendix A was performed to prepare plasmids and molecular clones. The chimeric molecular clone pMAL5UTR-NL4.3-Chimera-EGFP was made by replacing the 5′UTR (1–366) in the pNL4.3-CMV-EGFP [28] with the MAL 5′UTR (1–366). MutA mutation was introduced to the MAL-5′UTR plasmid and the pMAL5UTR-MutA-NL4.3-Chimera-EGFP using the primers listed in Appendix A. PBSM mutation was introduced to the NL4.3-5′UTR plasmid and the pNL4.3-CMV-EGFP in the same way. E478Q mutation was introduced to the RT expression plasmid pHXB2-RT, a kind gift from Dr. Donald Burke (University of Missouri, Columbia, MO, USA) [56,57], to generate pHXB2-RT-E478Q. All the plasmid sequences were confirmed by Sanger sequencing (DNA Core, University of Missouri, Columbia, MO, USA).

### 4.2. In Vitro RNA Transcription

RNA used for in vitro experiments were synthesized by T7 in vitro transcription. The DNA templates for RNA synthesis were produced by amplifying the corresponding sequences with a T7 promoter sequence. To determine optimal conditions for large-scale transcriptions, small-scale trial transcriptions were performed using varying concentrations of MgCl_2_. Large-scale transcriptions were carried out at volumes of 7.5–10 mL in transcription buffer (40 mM Tris-HCl, pH 8.0, 5 mM dithiothreitol (DTT), 10 mM spermidine, and 0.01% (*v/v*) Triton X-100) using dNTPs (12 mM each), Ribolock RNase Inhibitor (80 units, Thermo Fisher Scientific, Waltham, MA, USA), and T7 RNA polymerase (1 µM). Transcriptions reactions were held at 37 °C for 4 h and quenched with 1-M urea and 25 mM ethylenediaminetetraacetic acid (EDTA). RNAs were purified by sequencing gels, visualized by UV shadowing, electroeluted from the gel, and washed in ultra-centrifugal filters to remove acrylamide and salts.

### 4.3. HIV-1 NC, RT, and RT^E478Q^ Purification

Recombinant HXB2-RT and HXB2-RT-E478Q were expressed in *E. coli* BL21 (DE3)-pLysS cells (Invitrogen, Carlsbad, CA, USA) with 1-mM isopropyl β-D-1-thiogalactopyranoside (IPTG) for 4 h at 37°C, respectively. The harvested cell pellets were resuspended in lysis buffer (25 mM Tris, 500 mM NaCl, 0.15 mg/mL lysozyme, and 0.4-mM phenylmethylsulfonyl fluoride (PMSF), pH 8.0), sonicated, and centrifuged. The supernatant was applied to a cobalt column (HisPur Cobalt Resin, Thermo Fisher Scientific, Waltham, MA, USA) and purified according to the manufacturer’s protocols with an extra-high-salt buffer (25 mM Tris and 1 M NaCl, pH 8.0) washing step. The protein was then subjected to size exclusion chromatography for further purification and buffer exchange and stored in buffer containing 20-mM 4-(2-hydroxyethyl)-1-piperazineethanesulfonic acid (HEPES) (pH 7.5), 80 mM NaCl, and 2 mM tris(2-carboxyethyl)phosphine (TCEP) at −80 °C. The HIV-1 NC protein expression plasmid was a kind gift from Dr. Michael Summers (University of Maryland, Baltimore, MD, USA). The recombinant HIV-1 NC protein was expressed and purified exactly as previously described [58].

### 4.4. RNA Fluorescence Labeling

Cy3-labeled tRNA was prepared as described previously [59]. 3′-FTSC-labeled RNAs were prepared by incubating 20 µM of RNA in 200 µM of NaIO_4_ and 62.5 mM of NaOAc at room temperature in the dark for 90 min, followed by incubating with an additional 0.42 mM Na_2_SO_3_ at room temperature for 15 min and then mixing with 0.58 mM FTSC dye (dissolved in dimethylformamide) at 37 °C in the dark for 3 h. To quench the reaction, LiCl was added to a final concentration of 0.74 M, and the sample was mixed with 2.5 volumes of 100% ethanol and incubated at −80 °C for 1 h. The sample was centrifuged at 13,000× *g* for 20 min at 4°C, and the supernatant was discarded. The precipitated RNA was washed twice with 75% ethanol and lyophilized for 10 min. The lyophilized RNA was dissolved in double-distilled H_2_O and stored for further use.

### 4.5. NMR Experiments

RNA used for 1D ^1^H NMR experiments was prepared in 50–200 µM and dissolved in a ^2^H_2_O buffer containing 10 mM deuterated Tris-HCl (pH 8.0) and 2 mM MgCl_2_. RNA used for 2D ^1^H-^1^H Nuclear Overhauser Effect Spectroscopy (NOESY) experiments were prepared at 200–300 µM in the same buffer. The NMR data were collected at 308 K on a Bruker Avance III 800 MHz spectrometer equipped with a TCI cryoprobe. Data was processed with NMRPipe and NMRDraw using the NMRBox cloud-based virtual machine [60]. Data was analyzed using NMRViewJ (NMRFX).

### 4.6. RNA Annealing

Heat annealing was carried out by mixing HIV-1 vRNA with tRNA^Lys3^ at a 1:1 ratio in the annealing buffer (10 mM Tris-HCl, pH 7.5, 10 mM NaCl, 1 mM MgCl_2_, and 140 mM KCl) and incubated at 94 °C for 5 min, 85 °C for 15 min, 75 °C for 15 min, and 65 °C for 60 min. NC annealing was carried out by mixing HIV-1 vRNA with tRNA^Lys3^ at a 1:1 ratio and mixing with various amounts of NC, as indicated in the Results section. NC was then removed from the RNA complex by high-salt washes with 10 mM Tris-HCl, pH 7.5, 1.5 M NaCl, and 1 mM MgCl_2_) for the in vitro analysis.

### 4.7. Microscale Thermophoresis

3′-FTSC-labeled WT and mutant MAL 5′UTR RNA samples were pre-annealed to tRNA^Lys3^, and 50 nM of the complex was titrated with RT at concentrations varying from 0.00252 µM to 20.6 µM for the MAL 5′UTR:tRNA^Lys3^ experiments and 0.000763 µM to 6.2 µM for the MAL 5′UTR-MutA:tRNA^Lys3^ experiments. Experiments were performed using a Nanotemper Monolith and were analyzed using MO.Affinity Analysis (Nanotemper Technologies, Munich, Germany). Each experiment was repeated three times.

3′-FTSC-labeled WT and mutant NL4.3 RNA were annealed to tRNA^Lys3^ in annealing buffer using NC (vRNA: tRNA: NC = 1:1:12). NC was removed by high salt washes, as described in Section 4.6. The complexes were held at 50 nM in the microscale thermophoresis experiments, and RT was varied from 0.00198 µM to 65 µM for the PBS NL4.3: tRNA^Lys3^ experiments and 0.000763 µM to 25 µM for the PBSm-NL4.3: tRNA^Lys3^ experiments. Experiments were performed using a Nanotemper Monolith and were analyzed using MO.Affinity Analysis (Nanotemper Technologies, Munich, Germany). Three independent replicates of the measurements were performed for the K_d_ quantification. For the biphasic binding curve, K_d_ was determined by excluding data points for the low-affinity binding mode during the analysis of the high-affinity binding mode and vice versa [61].

### 4.8. Primer Extension Assay

The MAL and NL4.3 vRNA were annealed to 3′-Cy3 tRNA^Lys3^ in an annealing buffer by heat and by NC, respectively. The annealed complexes were mixed to 90 nM with 3 mM MgCl_2_, 1 µM NC, Ribolock RNase Inhibitor (Thermo Fisher Scientific, Waltham, MA, USA) 0.6 units, 2.7 µM RT, 25 mM HEPES, and 80 mM NaCl. To initiate the reaction, dNTPs were added to 90 µM and held at 37 °C. Each time point was quenched with 20 mM EDTA and placed on ice. SDS was added to 12.5 mg/mL, and samples were boiled for 5 min. Proteins were removed by phenol–chloroform extraction, and the RNA samples were boiled for 3 min and loaded onto an 8% denaturing 8 M Urea polyacrylamide gel. The gels were visualized by fluorescent scanning on a Typhoon FLA 9000 (GE Healthcare, Marlborough, MA, USA).

### 4.9. In Vitro Degradation Assay

The 3′-FTSC-labeled HIV-1 vRNA was pre-annealed to tRNA^Lys3^, as described in Section 4.6. In a 20 µL reaction, 166 nM of the RNA complex was mixed with 1.2 µM NC and 3 µM RT or RT^E478Q^ in a buffer containing 25 mM HEPES, 80 mM NaCl, and 0.6 units of Ribolock RNase Inhibitor (Thermo-Fisher Scientific, Waltham, MA, USA). The reactions were incubated at 37 °C for the specified times and quenched by the addition of EDTA. SDS was added to 12.5 mg/mL, and samples were boiled for 5 min. Proteins were removed by phenol–chloroform extraction, and the RNA samples were boiled for 3 min and loaded onto an 8% denaturing 8 M Urea polyacrylamide gel. The gels were visualized by fluorescent scanning on a Typhoon FLA 9000 (GE Healthcare, Marlborough, MA, USA).

### 4.10. Cells and Viruses

The 293FT cells were propagated in Dulbecco’s modified Eagle’s medium (DMEM, Sigma, St.Louis, MO, USA) supplemented with 10% fetal bovine serum (FBS). WT and mutant MAL chimeric viruses were produced by transfection [28]. Briefly, 293FT cells were co-transfected with 500 ng of WT and mutant MAL molecular clones (pMAL5UTR-NL4.3-Chimera-EGFP and pMAL5UTR-MutA-NL4.3-Chimera-EGFP) and 100 ng VSV-G (AIDS Reagent Program). The cells were rinsed and cultured in 10% FBS DMEM 6 h later, and the pseudo-typed viruses were harvested 48 h post-transfection. The cells were fixed with 4% paraformaldehyde, and the transfection efficiency was examined by using flow cytometry (Accuri Flow Cytometer, BD Biosciences, San Jose, CA, USA). The WT and mutant NL4.3 pseudo-typed viruses were produced in the same way using pNL4.3-CMV-EGFP and pNL4.3-PBSm-CMV-EGFP plasmids.

### 4.11. Infectivity Assay

The amount of pseudo-typed viruses was quantified by p24 ELISA, using precoated MicroFleur well plates (Optofluidic Bioassay, Ann Arbor, MI, USA). Pseudo-typed viruses containing 250 ng of Gag p24 was used to infect 2 × 10^5^ of TZM-bl cells (AIDS Reagent Program) in 12-well plates. The infected cells were harvested 20 h post-infection and fixed with 4% paraformaldehyde. The percentage of EGFP cells were measured by Accuri Flow cytometer.

### 4.12. Quantification of cDNA Products in Infected Cells

Equivalent amounts (250 ng of Gag p24) of WT and mutant pseudo-typed viruses were pretreated with DNase I and DpnI and then infected 2 × 10^5^ of 293 cells in 12-well plates. The cells were harvested 2 h post-infection and subjected to DNA extraction with QIAamp DNA Blood Mini Kit (Qiagen, Hilden, Germany). Cellular DNA (2 µL) was evaluated by qPCR using primer sets specifically targeting the reverse transcription products [29]. The WT and mutant qPCR data were normalized by the respective cellular DNA concentrations.

### 4.13. RNA Integrity Assay

A total of 250 µl of pseudo-typed viruses were used for RNA extraction with TRIzol Reagent (Invitrogen Carlsbad, CA, USA) following the manufacturer’s user guide. The extracted nucleic acids were subjected to DNase I and DpnI digestion (NEB, Ipswitch, MA, USA). The viral genomic RNA was extracted using a NEBNext Poly(A) mRNA Magnetic Isolation Module (NEB, Ipswitch, MA, USA), and the integrity were evaluated by qRT-PCR using the primer and probe sets listed in Appendix A. The RT-qPCR was performed with the CFX96 Real-Time PCR Detection System (Bio-Rad, Hercules, CA, USA).

### 4.14. Statistics Analysis

An unpaired *t*-test was used to compare the significance of differences between different groups with Prism (GraphPad Software, San Diego, CA, USA). The *p*-values less than 0.05 were considered significant. Statistical differences are indicated as follows: * *p* < 0.05, ** *p* < 0.01, *** *p* < 0.001, and **** *p* < 0.0001. More than three biological replicates were applied in all the experiments.

## Figures and Tables

**Figure 1 ijms-22-00058-f001:**
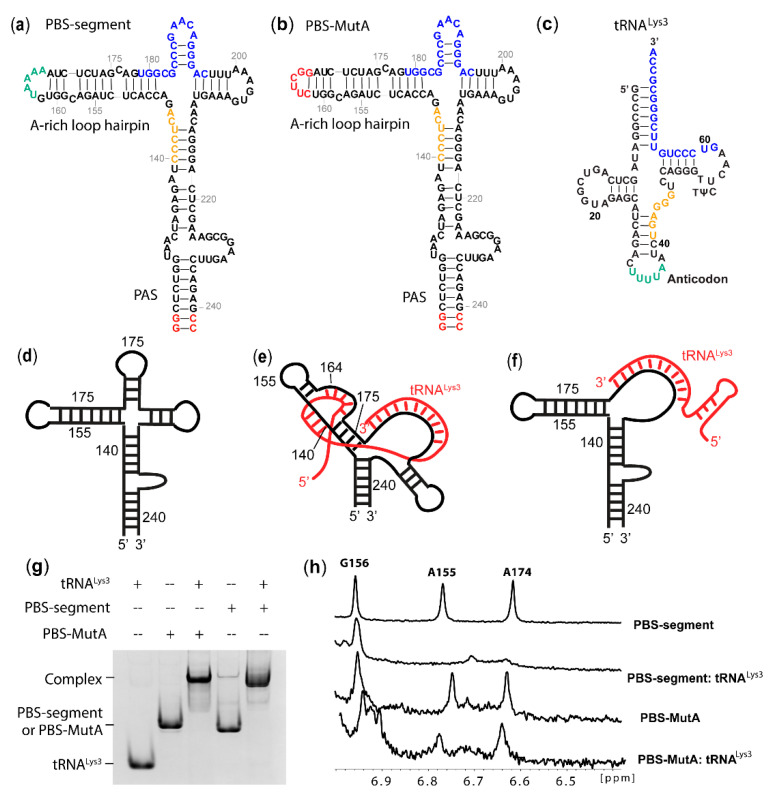
Mutation of the A-rich did not abolish tRNA^Lys3^ annealing but disrupted the possible A-rich loop: anticodon interactions. (**a**) Predicted secondary structure of protein binding site (PBS) segment RNA. The A-rich loop residues complementary to the anticodon of tRNA^Lys3^ are labeled in green, with additional complementary residues labeled in yellow. The 18-nt PBS is labeled in blue. Two non-native G-C pairs were added (red) to enhance the T7 transcription. (**b**) Predicted secondary structure of PBS-MutA, of which G162-A167 were substituted by CUUCGG (red). (**c**) Secondary structure of tRNA^Lys3^. The 3′-18-nt residues complementary to PBS are labeled in blue. Anticodon loop residues that are complementary to the PBS segment A-rich loop are labeled in green, with additional complementary residues labeled in yellow. (**d**–**f**) Cartoon models of the PBS segment (**d**), PBS segment: tRNA^Lys3^ complex with extended A-rich loop: anticodon intermolecular interactions (**e**), and without the extended interactions (**f**). The MAL PBS segment is labeled in black, and tRNA^Lys3^ is labeled in red. (**g**) Mutation of the A-rich loop did not prevent tRNA^Lys3^ annealing. Complex formation was examined using an 8% native PAGE. (**h**) Mutation of the A-rich loop disrupted the possible A-rich loop: anticodon interactions. Portion of 1D ^1^H NMR of the PBS segment and PBS segment: tRNA^Lys3^ complexes (top) and PBS-MutA and tRNA^Lys3^ complexes (bottom). The signal-to-noise of the PBS-MutA samples were lower, because the RNA sample concentrations used for NMR data collection were lower than the wild-type (WT) RNAs.

**Figure 2 ijms-22-00058-f002:**
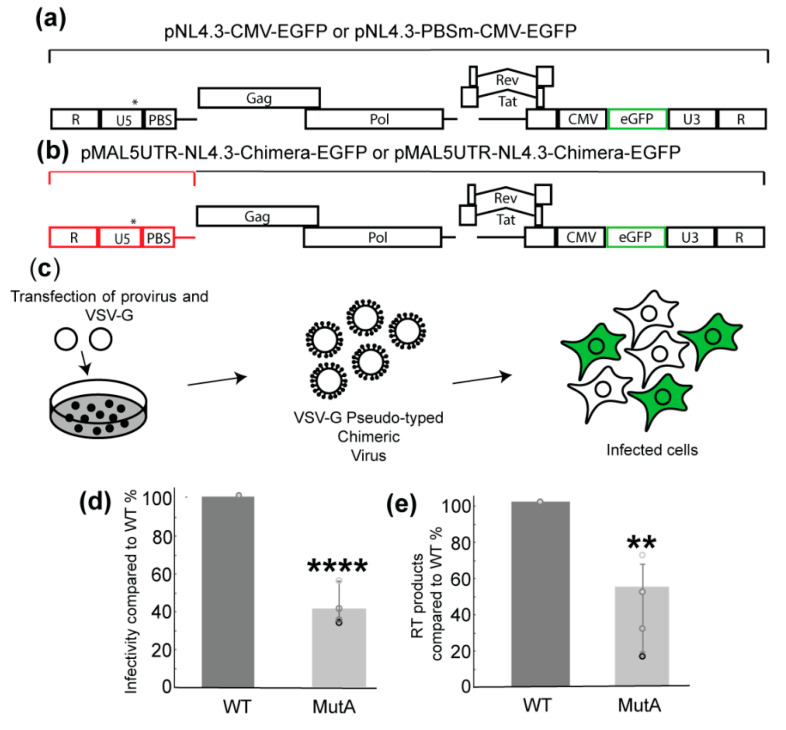
Cell-based infectivity and reverse transcription product assays to compare the replication efficiency of the WT and Mut-A viruses. (**a**) Map of the pro-viral plasmid used for the pseudo-typed virus production and cell-based assays. An asterisk indicates the approximate location of A-rich loop mutations. (**b**) A chimeric molecular clone was created using the MAL 5′UTR and NL4.3 backbone. The MAL 5′UTR is highlighted in red. An asterisk indicates the approximate location of A-rich loop mutations. (**c**) Schematic of a single-cycle infectivity assay. Pro-viral and vesicular stomatitis virus (VSV-G) plasmids were transfected in 293FT cells. VSV-G pseudo-typed chimeric viruses were harvested and used to infect TZM-bl cells. Flow cytometry was used to measure the percentage of EGFP-expressing cells. (**d**) Mutation of the A-rich loop lowered the infectivity of the pseudo-typed virus. (**e**) Mutation of the A-rich loop led to reduced reverse transcription products in cells infected by the pseudo-typed virus. The cDNA products were quantified by qPCR. Graphs are representative of four experiments, average ± standard deviation; ** *p* < 0.01 and **** *p* < 0.0001.

**Figure 3 ijms-22-00058-f003:**
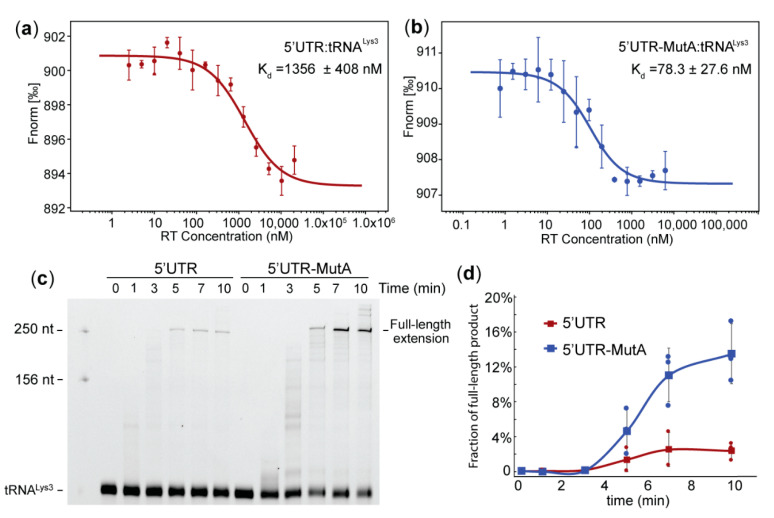
Mutation of the A-Rich loop led to increased affinity for reverse transcriptase (RT) binding to the vRNA: tRNA^Lys3^ complex and increased primer extension efficiency in vitro. (**a**) microscale thermophoresis (MST)-binding curve of RT binding to the 5′UTR: tRNA^Lys3^ complex. (**b**) MST-binding curve of RT binding to the 5′UTR-MutA: tRNA^Lys3^ complex. Representative curves of three independent experiments are shown in (**a**,**b**), average ± standard deviation. (**c**) Mutation of the A-rich loop in the template RNA led to an increased primer extension. The primer extension products of Cy3-labeled tRNA^Lys3^ on the 5′UTR WT (left) and 5′UTR-MutA (right) were visualized by polyacrylamide gel electrophoresis and imaged by fluorescent scanning. (**d**) Quantification of the full-length primer extension products averaged from three independent experiments shows enhanced primer extension efficiency upon mutation of the A-rich loop.

**Figure 4 ijms-22-00058-f004:**
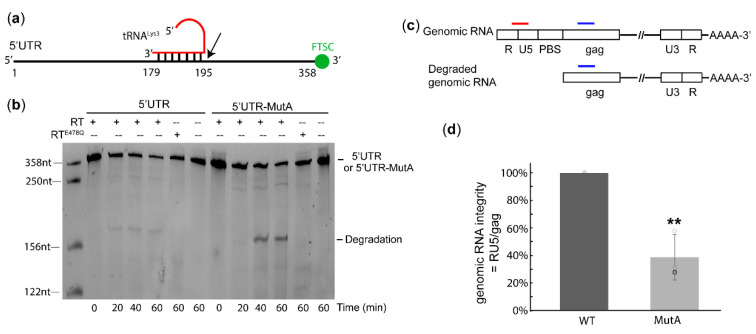
A-rich loop mutation in viral RNA (vRNA) resulted in accelerated RNA degradation in both in vitro and in cell-based assays. (**a**) Schematic diagram for the in vitro degradation assay. tRNA^Lys3^ was annealed to 3′-fluorescein thiosemicarbazide (FTSC)-labeled 5′UTR or 5′UTR-MutA. Black arrow marks the approximate RNaseH degradation site. (**b**) In vitro degradation assay of the 5′UTR: tRNA^Lys3^ and 5′UTR-MutA: tRNA^Lys3^ complexes shows enhanced degradation of the mutant complex. (**c**) Schematic diagram of cell-based genomic integrity assay. Primers targeting the RNA upstream and downstream of PBS are sketched in red and blue, respectively. (**d**) Genomic RNA integrity assay shows decreased genomic RNA integrity upon mutation of the A-rich loop. Graph is representative of three experiments, average ± standard deviation; ** *p* < 0.01.

**Figure 5 ijms-22-00058-f005:**
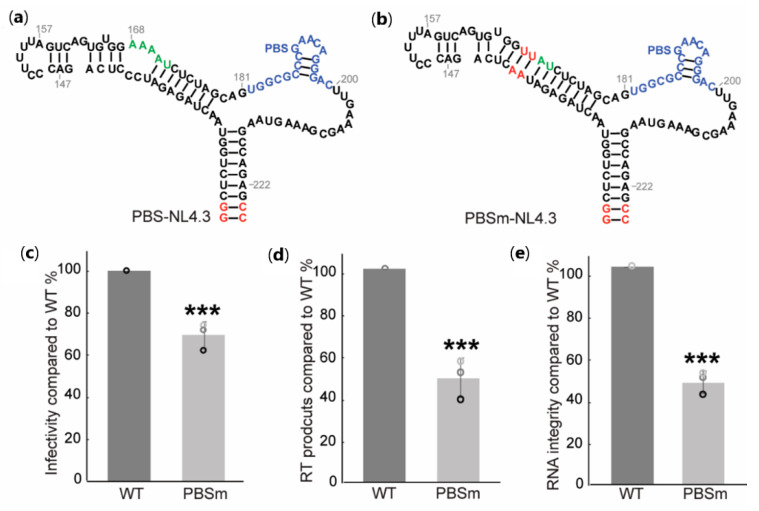
Cell-based assays show the A-rich loop residues are important for viral replication in human immunodeficiency virus 1 (HIV-1) subtype B. (**a**) Predicted secondary structure of PBS-NL4.3 RNA. The A-rich loop is labeled in green, the primer binding site is labeled in blue, and the non-native nucleotides are labeled in red. (**b**) Predicted secondary structure of PBSm-NL4.3 RNA. A PBS that is complementary to the 3′-18-nt of tRNA^Lys3^ is labeled in blue, and the non-native nucleotides are highlighted in red. (**c**) Mutation of A-rich loop lowered the infectivity of the pseudo-typed virus in the single-round infectivity assays. The infectivity was quantified by measuring the EGFP signals and normalizing the p24 levels of the virions, and the infectivity of the mutants was normalized to the WT in each parallel experiment. (**d**) Mutation of the A-rich loop reduced the reverse transcription products 2 h post-infection. (**e**) Mutation of the A-rich loop resulted in reduced vRNA integrity in the genomic integrity assay. Graphs are representative of three independent experiments, average ± standard deviation; *** *p* < 0.001.

**Figure 6 ijms-22-00058-f006:**
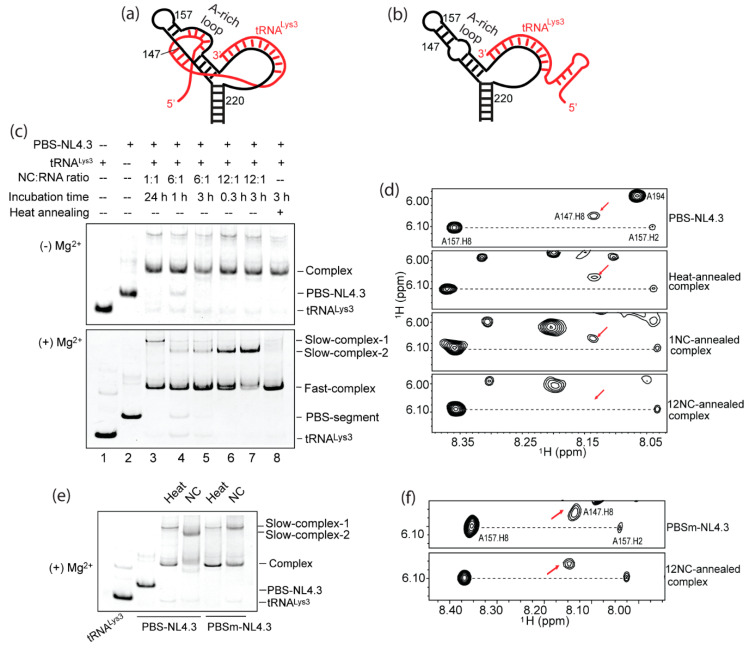
Native gel and NMR analysis of annealing conditions on PBS-NL4.3 and PBSm-NL4.3 show that the A-rich loop: anticodon interaction was promoted by nucleocapsid (NC) annealing but not by heat annealing. (**a**) Sketch of the A-rich loop: anticodon interaction model. PBS-NL4.3 is shown in black, and tRNA^Lys3^ is shown in red. (**b**) Sketch of the complex model without extended interactions. (**c**) Complexes annealed under various conditions adopt different structures. The PBS-NL4.3: tRNA^Lys3^ complexes were analyzed by polyacrylamide gel electrophoresis without Mg^2+^ (top panel) and with Mg^2+^ (bottom panel). (**d**) Portions of 2D ^1^H-^1^H NMR of the PBS-NL4.3: tRNA^Lys3^ complexes show that only the 12:1 NC: RNA complex led to a possible A-rich loop anticodon interaction. The red arrow indicates an A147.H8 chemical shift. (**e**) Native gel analysis of the complexes of PBS-NL4.3:tRNA^Lys3^ and PBSm-NL4.3: tRNA^Lys3^ annealed by heat and by 12 NC. (**f**) Portions of 2D ^1^H-^1^H NMR of PBSm-NL4.3 and the PBSm-NL4.3: tRNA^Lys3^ complex show that the mutation of the A-rich loop disrupted the formation of a possible A-rich loop anticodon interaction. The red arrow indicates an A147.H8 chemical shift.

**Figure 7 ijms-22-00058-f007:**
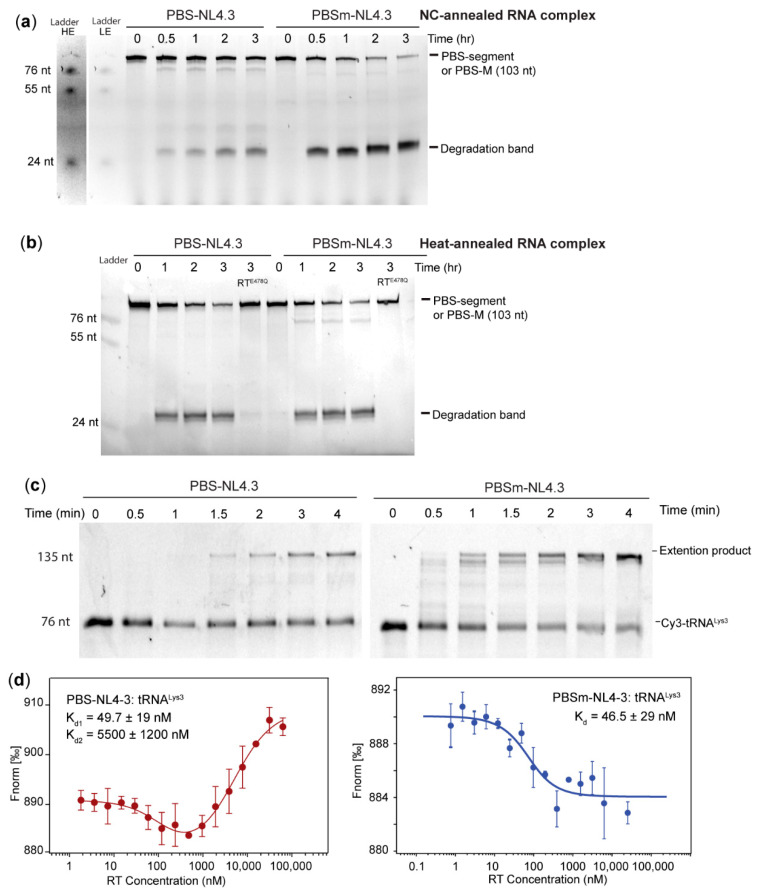
The A-rich loop: anticodon interaction is conserved in HIV-1 NL4.3 to protect the vRNA integrity. (**a**) The PBSm-NL4.3 RNA in the NC-annealed complex was degraded faster the WT RNA in the NC-annealed complex. Ladders are shown in contrasted adjusted levels (left panel, HE = high exposure and LE = low exposure). (**b**) The PBSm-NL4.3 RNA in the heat-annealed complex was degraded at a similar rate as the WT RNA in the heat-annealed complex. (**c**) Primer extension on the mutant PBSm-NL4.3: tRNA^Lys3^ complex was more efficient than on the WT complex. (**d**) MST binding curves of PBS-NL4.3: tRNA^Lys3^ (left panel) and PBSm-NL4.3: tRNA^Lys3^ (right panel) titrated with RT. Representative curve of three experiments is shown; average ± standard deviation.

## Data Availability

Please refer to suggested Data Availability Statements in section “MDPI Research Data Policies” at https://www.mdpi.com/ethics.

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
