# Peer review of "Extended Interactions between HIV-1 Viral RNA and tRNA^Lys3^ Are Important to Maintain Viral RNA Integrity"

_ijms, 2020, doi:10.3390/ijms22010058_

Round 1
Reviewer 1 Report
In the manuscript by Gremminger and colleagues describe extended vRNA: tRNAlys3 interactions between HIV-1 RNA and tRNAlys3. They first examine the HIV-1 Mal strain and followed by the HIV-1 NL4-3 strain. Specifically, the authors introduce mutations within the A-loop hairpin (PBS-MutA) and analyze interactions using NMR spectroscopy, reverse transcriptase binding, primer extension assays with reverse transcriptase, vRNA stability, and single cycle infectivity assays. Using the HIV Mal strain, the authors show using NMR that the A-loop residues are likely involved in extended vRNA: tRNAlys3 interactions, that there is increased RT binding to the template, enhanced primer extension efficiency but the vRNA was less stable in virions. Overall, this is a well written paper and would add new information to the field. I believe that the authors should carefully read the manuscript and correct typos and misspellings. My biggest concern is biological relevance to virus replication. My comments are below:
- The details of the infectivity assays are scant. These assays should be more detailed as they are critical to the overall conclusions of the paper. A schematic diagram of the viruses created would helpful to readers of the article.
- The authors should point out if the A-loop region is found in the different subtypes of HIV-1 and other primate lentiviruses.
- The authors show that the nucleocapsid protein (NC) was necessary for the formation of extended A-rich loop: anticodon interactions in vitro. Are these interactions also required for HIV-1 Mal? This should be presented.
- The authors show that A-loop mutations result in a decease in infectivity of approximately 50% in single cycle infectivity assays. While this may be statistically significant it is likely not biologically significant. A better way to show significance would be to introduce the A-loop mutations into a replication competent genome (i.e., pNL4-3, which is available from the NIH AIDS Reagents program) and analyze infectivity in a spreading infection. After several rounds of infection/reinfection the reduction in infectivity may be more pronounced. Further, the authors could determine if reversions and/or compensating mutations occur that enhance replication.
Reviewer 2 Report
I fould several, trivial misarrangements of figures and text that should be checked by authors and be corrected if necessary. And also, I have one minor question to be answered.
Figure 1, (a) and (b)
It seems that U139 should be black, not yellow, if the yellow sequence should be complementary to the yellow sequence in (c).
Figure 1, (h)
The peak labels, G175, A174, and A155 may be G156, A155, A174, respectively, if the assignments in Figure S1 (c) are correct.
Line 256
infactivity -> infectivity
Lines 331 and 340
A-rich:loop:anticodon -> A-rich loop:anticodon
Line 335
completed -> completely (?)
Line 479
Ecoli -> E. coli (in italics)
Line 489
in vitro -> (to be italicized)
Minor question:
Figure 6(c), the lower panel (+Mg):
The "First-complex" band seems to contain two distinct bands. What is the authors' explanation about the result?
It seems to me that the heat annealed complex appears as the lower fast-complex band. The 12NC-annealed sample may first form an intermediate that appears as the upper fast-complex band and, then, forms slow-complex-2 slowly. Is this right?
In other words, the fast-complex in lane 6 has two different forms: the lower band is the same as the fast-complex in lane 8, and fast-complex in lane 7 contains mainly the upper species.
